# OpenReview forum: "KDA: A Knowledge-Distilled Attacker for Scalable LLM Red Teaming"
_ICLR.cc/2025/Conference — ICLR 2025 Conference Withdrawn Submission_

### Official Review · Reviewer_JAXP · 2024-10-30

**Soundness:** 2
**Presentation:** 2
**Contribution:** 3
**Rating:** 1
**Confidence:** 4

**Summary:**

This paper introduces KDA (Knowledge-Distilled Attacker) for automatically generating malicious prompts for LLMs. They claim to be the first attack which satisfies the ABCDE (Automatic, Black-Box, Coherent, Diverse, and Exempt) properties desirable for LLM jailbreaking. The paper claims that KDA is both more successful and faster than existing methods. They further introduce the RedTeam-10k dataset which contains 10k harmful attack prompts spanning 12 different categories. Finally, they train a human-aligned Judge (HJ) LLM to determine whether the attack was successful. The paper claims that HJ reduces false negatives while maintaining high precision compared to Text Matching, GPT-4, and LLama-Guard-2.

**Strengths:**

The paper poses the idea of finetuning an LLM on existing malicious prompts in order to learn to generate malicious prompts given a target behaviour, which to the best of my knowledge is a novel idea. According to the results KDA is highly successful at jailbreaking achieving 100% attack success rate across all models and harmful queries.

The paper is relatively well written, structured, and clear (with some exceptions listed below). If my concerns are addressed I would be happy to accept this paper just based on the strength of the results and the contribution of an attack method, evaluator, and dataset.

**Weaknesses:**

While I believe this paper and its ideas could be a valuable addition to the current literature, I think there are a number of major issues that drive me towards rejection.

**Claims**

In my opinion, the most glaring weakness of this paper is its claim that KDA satisfies ABCDE properties. The main issue is that KDA is trained on and in large part relies on attacks (AutoDAN, GPTFuzzer, PAIR) which the paper itself claims do not satisfy ABCDE. In particular, E is very important as being exempt from commercial LLMs is an important claim as it lists a large number of techniques which already satisfy ABCD but not E. In order to train KDA one must first generate attacks from all of the $N_{att}$ SOTA attack methods. Therefore, I would argue that if any of these methods uses a commercial LLM then you must also claim that KDA is not E. This in someway also challenges the claim of B as AutoDAN is not B. One could claim that the KDA technique is general and could be used with only E SOTA attackers; however, one could make a similar argument for AutoDAN, GPTFuzzer, and PAIR which to the best of my knowledge also are general and have no requirement on using commercial LLMs (although they do in their evaluation). So by this argument, AutoDAN, GPTFuzzer, and PAIR should be marked as E and the claim that KDA is the first method to satisfy all 5 ABCDE properties is no longer true.

The paper also claims KDA is coherent and diverse; however, the evaluation makes no effort into showing this. While given the method I am inclined to believe this claim, the paper needs to address these claims with experiments (i.e. showing some perplexity metric and/or giving some metric on diversity).

**Evaluation**

The evaluation seems incomplete and/or is poorly written. Some evaluation details are unclear, like what test set is being used to evaluate? The paper seems to indicate that there are 2000 data samples split into 60/20/20 train/validate/test. So are you evaluating the human judge and KDA on this test set of size 400? If so, in Table 2 how is it possible to get an accuracy of 87.33 or 88.67 (i.e. 364/400 = 0.885, 365/400 = 0.8875)?

Algorithm 1 says that they sample K attack prompts then check the response from the target and Section 5.3 seems to indicate that an average of just over 1 query is needed. This suggests that a majority of the time the first generated attack works. Which attack is that? Generated in the style of which attack name (see line 290)? If a majority of the time the first attack works, why do we need to train on multiple attack methods, is one enough?

There seems to be a lot of "cross pollination" between the training of KDA, HJ and the evaluation. There are 3 other attack methods chosen: AutoDAN, GPTFuzzer, and PAIR (AGP). KDA in the evaluation is trained on AGP, the HJ is (seemingly, it is unclear) trained on the same data generated by AGP, KDA is evaluated against AGP and HJ is evaluated against competitors on a dataset generated by AGP.

There are missing evaluations. As mentioned above, it needs to be shown that KDA is C, D. I think it would also be good to show how KDA performs against LLM defenses (like AutoDAN, Page 7 Effectiveness against defense). How does KDA perform against methods it is not trained on?

After claiming that KDA does not rely on commercial LLMs, the harmful dataset construction is done with multiple commercial LLMs (i.e. GPT-4, GPT-3.5). This harmful dataset goes on to be the foundation for the training data which KDA is trained on and which the HJ evaluator is trained on. Many of the choices made are unexplained/unclear, i.e. why are 200 instances of Harmful-1k sampled when all samples could be used? $N_{att} = 3$, there are 200 queries, and 2 target models. How does this become 2000 data samples? Additional questions like this included below.

**Clarity**

Along with the issues above, there are some other issues with clarity.

Line 226 gives a definition of an optimization problem, then claims that many attack methods approximate it but has shortcomings. There is no where in the paper that I see which then addresses the optimization problem. Does KDA approximate it? Additionally, are you maximizing over all $A$s? Is there some constraint on the length of $A$? Is there some penalty for being coherent?

In line 221 $R_A \sim q_T(\cdot|A)$ where A is an attack prompt. However, in line 300, KDA is trained by minimizing the cross entropy between $q_{KDA}(\cdot|X)$ and target prompt $A$ where $X$ is a prompt and also a function? There is some mismatch in the notation. Detailed questions listed below.

**Structure**

Given my comments above the paper is poorly structured. Currently, too much of the paper talks about related works and there is not enough space to clearly describe KDA or give adequate experiments. There are also a large number of figures which take up considerable space which I think could be combined or are not needed.

**Questions:**

I am willing to increase my score if the below questions are adequately addressed.

**Claims**

1. Is KDA E if it uses commercial LLMs to generate the training data and if it depends on methods which it claims are not E (i.e. if commercial LLMs are used to learn KDA then doesn't KDA depend on commercial LLMs)?
2. If KDA is E because it could use non-commercial LLMs to generate data or use E SOTA attack methods, then shouldn't many of the previous works also be considered E? Then it seems less clear that KDA is the first ABCDE attack method.
3. Can you show that KDA is coherent and diverse?

**Evaluation**

4. Which attack methods are sampled first from $q_{KDA}$? Is it random? If not, it seems like a single method is basically good enough given Section 5.3 or can you show in an ablation study what happens if you use $N_{att} =1,2$?
5. (line 321) Why do you use GPT-4 to construct malicious queries and GPT-3.5 to generate target strings?
6. (line 322) What does it mean to have the judge LLM filter out queries consistently refused by Llama, etc? To me 'filter out' implies that these queries are getting removed, but I don't think that's what you mean.
7. (line 322) If you have 12 categories, how do you end up with 1000 queries? The categories must be mismatched
8. What test set is being used, see the weakness section, the paper indicates a test set size of 400, but the data does not seem to line up with that.
9. Figure 7 evaluates across the 12 categories, what dataset is this? Is it the 400 test set? If so how is this split between the classes?
10. Is it fair to compare HJ to TM, GPT-4, LG-2 on the KDA dataset? HJ is trained on that data, none of the other methods are. How does HJ compare on other datasets?
11. How does KDA compare against other methods it is not trained on?
12. How does KDA perform against LLM defenses?
13. (line 327) Why are 200 instances chosen out of 1000?
14. (line 342) Same 200 instances as above?
15. (line 329) How do you get 2000 data samples from 200 instances, 2 target models, and 3 attack methods? Shouldn't it at least be divisible by 3? If you generate 1 each then shouldn't your dataset be 200x2x3 = 1200 data samples?
16. (line 405) Is the KDA training set only different from the human-annotated dataset because it doesn't have human labels?
17. Can we see an attack generated by KDA?

**Clarity**

18. (line 226) What is the point of the optimization problem?
19. Does KDA approximate Equation 1? Shouldn't there be some additional constraints on $A$? Doesn't $A$ depend on $Q$?
20. (line 221) What is $X$ a function or a prompt? What is the $\cdot$ in $q(\cdot|A)$?
21. (line 475) KDA does require additional queries of LLMs, it requires one query to the attack generation LLM and one query to the judge LLM.
22. How does KDA's response change for different LLMs (I guess the paper makes it seem like it does not) so if AutoDAN/GPTFuzzer/PAIR style attacks do not work on a given LLM it seems that KDA may just endlessly generate new random prompts in these styles without ever stopping.
23. Does KDA change its response given different queries? Does it have one "universal" jailbreak that it just inserts the query into? Can you take a jailbreak for one query and insert another query?

**Corrections not affecting my score**

1. Line 197 - AutoDAN avoids these dependencies? or doesn't (Line 189)?
2. Line 449 - Guard is misspelled
3. Figure 3 - Figure calls dataset Harmful-10k however everywhere else it is referred to as Harmful-1k
4. Does the A5000 GPU have 24.5GB of memory? I only see 24 GB online although I have not used this GPU personally
5. Line 407 - extra period
6. Line 537 - KDA is unable to jailbreak an LLM if no attack prompts exist? Is this really a weakness? If it is a perfectly robust LLM it seems obvious that this is the case, or are you saying that KDA wouldn't work on LLMs whose attack prompts it is not trained on because your results against GPT-4 seem to contradict that?
7. RedTeam-10k can be used for industrial-scale red teaming? Vicuna seems like the weakest model you test against and it is a single model.

---

### Official Review · Reviewer_JGC5 · 2024-11-01

**Soundness:** 2
**Presentation:** 3
**Contribution:** 2
**Rating:** 3
**Confidence:** 4

**Summary:**

This paper proposes a new jailbreaking attack against black-box LLMs. Technically, the paper finetunes another LLM to generate attack prompts against a target LLM. Together with the proposed method, the paper also releases a dataset of attack prompts that can be used for large-scale jailbreaking tests.

**Strengths:**

+ The paper gives a relatively comprehensive summary of existing works, analyzing their pros and cons.

+ The paper releases a dataset that is valuable for LLM red-teaming testing and defense training.

**Weaknesses:**

- Although most criticisms of existing works are accurate, the last one ``Lack of human-aligned jailbreak evaluator'' is vague. First, it is not clear how to define and evaluate human-aligned jailbreaking evaluator.  For example, GPTFUZZER's judgment model is trained from a dataset with human annotations. Given that the labels are assigned by humans and the classifier has high accuracy, we can also state that it is aligned with humans. I am not sure why this work marks it as not aligned with humans.

- The proposed method is super clear to me. For example, when generating the fine-tuning dataset what $\mathcal{M}$ refers to, what are the attack methods used in this work? Also, what is the fine-tuning objective function? Next token prediction loss or SFT?

- The proposed method lacks technical depth and novelty. Directly fine-tuning another model to generate attack prompts is not new. [1] proposes a similar idea already. In [1], they consider more regularization terms to ensure the diversity of the generated prompts. Also, it is not clear to me why using the knowledge distillation method rather than using RLHF or DPO to train the model generates attack prompts against one or multiple target models. Such a method should be more flexible than distilling a given dataset because the model has the freedom to explore different attack generation strategies. Also, why not directly use the dataset created for the jailbreaking attacks, what are the main benefits of the distillation other than efficiency?

-  The judgment model mainly focuses on whether the model output is harmful. It cannot decide whether the output is related to the input query. As discussed in [2], this will introduce false positives.

- The jailbreaking performance of the proposed method is similar to GPTFUZZER from Fig. 7. I would suggest authors provide a more details explanation of the results and better justify the benefit of their approach compared to GPTFUZZER.

- I would suggest the authors explore simple defenses given their dataset. They can try to train a simple guardrail model to see if it helps filter adversarial outputs.

[1] AdvPrompter: Fast Adaptive Adversarial Prompting for LLMs
[2] When LLM Meets DRL: Advancing Jailbreaking Efficiency via DRL-guided Search

**Questions:**

See the above weaknesses for my questions.

---

### Official Review · Reviewer_DZad · 2024-11-04

**Soundness:** 2
**Presentation:** 3
**Contribution:** 2
**Rating:** 3
**Confidence:** 4

**Summary:**

This paper proposes a Knowledge-Distilled Attacker (KDA) to address limitations in existing jailbreak methods for LLMs. KDA leverages a knowledge distillation approach to generate attack prompts that bypass LLM safeguards, achieving scalability and efficiency by training on a diverse set of jailbreak prompts. Key contributions include (1) a human-aligned jailbreak evaluator for effective success measurement, (2) the KDA model that operates in a black-box, coherent, and diverse manner, and (3) the introduction of the RedTeam-10k dataset with 10,000 harmful prompts. The authors demonstrate KDA's effectiveness through empirical evaluations, showing high attack success rates and reduced attack generation time.

**Strengths:**

The paper presents a straightforward method, along with a large dataset that could greatly benefit evaluations within the community.

**Weaknesses:**

Lack of contributions. The paper claims three main contributions: a human-aligned evaluator, an attacker LLM model with its training design, and a jailbreak dataset. However, each of these contributions has existing counterparts in prior work, and the paper does not clearly demonstrate its advantages or uniqueness compared to these studies. Specifically:
1. Human-Aligned Evaluator: Previous works have proposed fine-grained jailbreak evaluators, such as those in Harmbench [1] and StrongReject [2]. The paper, however, only compares its evaluator with naive methods (e.g., text matching), a vanilla LLM (GPT-4), and defense guardrails (Llama-Guard-2). None of these baselines are specifically designed for jailbreak evaluation. Therefore, the advantages or unique features of the proposed human-aligned evaluator over existing methods remain unclear.
2. LLM Attacker: Similar methodologies have been employed in prior studies. For example, AmpleGCG [3] trains an attacker LLM using GCG [4] attack data, and AdvPrompter [5] trains an attacker LLM with AutoDAN [6] data. This indicates that training an LLM to generate jailbreak prompts is not a novel idea. Unfortunately, the paper does not include comparisons with these methods. It seems that the main difference is the utilization of more jailbreak methods for generating prompts during training. If so, the contribution may be limited.
3. Generated Dataset: While contributing datasets to the community is commendable, the proposed dataset is generated by an attacker model trained on existing attacks like GCG and AutoDAN. This raises the question: why not directly use the jailbreak data from these existing attack methods, such as the data generated by Harmbench [1], which includes a more diverse set of jailbreak methods? The uniqueness and added value of the proposed dataset are not evident, potentially diminishing the significance of this contribution.

[1] Mazeika, Mantas, et al. "Harmbench: A standardized evaluation framework for automated red teaming and robust refusal." arXiv preprint arXiv:2402.04249 (2024). \
[2] Souly, Alexandra, et al. "A strongreject for empty jailbreaks." arXiv preprint arXiv:2402.10260 (2024). \
[3] Liao, Zeyi, and Huan Sun. "Amplegcg: Learning a universal and transferable generative model of adversarial suffixes for jailbreaking both open and closed llms." arXiv preprint arXiv:2404.07921 (2024).
[4] Zou, Andy, et al. "Universal and transferable adversarial attacks on aligned language models." arXiv preprint arXiv:2307.15043 (2023). \
[5] Paulus, Anselm, et al. "Advprompter: Fast adaptive adversarial prompting for llms." arXiv preprint arXiv:2404.16873 (2024). \
[6] Zhu, Sicheng, et al. "Autodan: Automatic and interpretable adversarial attacks on large language models." arXiv preprint arXiv:2310.15140 (2023).

**Questions:**

Please refer to the weaknesses. Additionally:
1. If I understand correctly, the KDA model is trained on jailbreak prompts that successfully attack the target model, and then evaluated on these same models. If this is accurate, it raises fairness concerns, as KDA learns from successful attacks during training, unlike other baselines that do not rely on such data. Could the authors provide insights into KDA's ability to generalize to models not explicitly covered in the training set? Additional analysis on generalizability across diverse model architectures would help strengthen confidence in KDA's robustness.

---

### Official Review · Reviewer_gmui · 2024-11-04

**Soundness:** 2
**Presentation:** 3
**Contribution:** 2
**Rating:** 3
**Confidence:** 2

**Summary:**

This paper is targeting the research gap of developing a scalable jailbreak method for red teaming LLMs that possesses all the “ABCDE properties” – ie. the attack method is Automatic, Black-box, Coherent, Diverse, and Exempt from dependency on commercial LLMs.
The paper's contributions include:
* A human-aligned jailbreak Evaluator: to create this they finetune Llama-Guard-2 in a supervised manner on ~1K manually annotated LLM responses across different attack methods to create a human-aligned judge LLM that can identify successful jailbreaks.
* The Knowledge-Distilled Attacker (KDA): created by fine tuning the open source LLM Vicuna-13B on a set of attack prompts (~1K) generated by 3 SOTA attack methods: AutoDAN, PAIR and GPTFuzzer. This creates a framework to automatically generate black-box attack prompts with a low attack generation time of under 10 seconds per attack.
* A large-scale attack dataset: they develop the RedTeam-10k dataset, including the attack prompts.

They evaluate KDA against 3 benchmarks SOTA attackers on 2 open source and 2 closed source LLMs, where KDA achieves a perfect attack success rate with significantly less time per success.

**Strengths:**

* This is an Important topic.  The paper is relatively easy to understand, with the figures greatly enhance readability.
* Their approach leverages the strengths of three attack methods: AutoDAN, GPT-Fuzzer and PAIR -- chosen for their diverse attack strategies, each addressing aspects of their desired “ABCDE properties”.
* Experiments demonstrate strong attack success rate on both the open and closed-source LLMs analysed, while their approach doesn't have any commercial LLM dependency for attack generation.
* Experiments demonstrate genuine significant reduction in time per success. Low time per success is maintained even in categories that other LLMs have lower ASR on.

**Weaknesses:**

* As noted by the authors, its effectiveness heavily relies on the presence of successful attack prompts.

* The improvement achieved by the Judge/Evaluator over basic text matching is marginal, considering the substantial effort required for human labeling and fine-tuning the LLM.

* Lack of comprehensive evaluation:
  i) The paper hasn’t tested generalizability to other benchmark datasets outside of Harmful-1K, which was used to train KDA.
  ii) Jailbreak performance in the main paper has only been compared with the 3 attack models that KDA leveraged in its training.
  iii) Similarly, jailbreak performance in the main paper has only been compared with the 3 target LLM models that successful attack prompts were collected from and then KDA was finetuned with. In Appendix A.1 -- the transfer attack is only performed on 100 samples and it is using an LLM from the same LLM model family that KDA was fine-tuned with. (eg. KDA has been fine-tuned with successful GPT-4 attacks and then this transfer evaluation is performed on GPT-4o which is the same model family).
  iv) Results from Table 2, Figure 6 and Figure 7 are from small test sets (~100 samples?). The jailbreak performance comparison in particular should be carried out on multiple datasets, including larger ones. (The large scale evaluation in Table 3 is only targeting Vicuna-7B, which all 4 models had a high ASR on in Figure 6, so this doesn't add to the jailbreak performance evaluation.).
   v) the number of queries per success not shown for the jailbreak performance comparison.

* No mention of ethics review or ethics board approval being obtained for this work.

**Typos:**
Bottom of page 9: “KDA achieves a perfect ASR while being at least 4 times faster than PAIR, 7 times faster than PAIR,…” one of the PAIRs should be GPTFuzz.

**Questions:**

1. Please make the evaluations more extensive by:
* Showing jailbreak performance comparison on a large dataset that has not been used in the training of the framework - eg. HarmBench (ICLR '24). In this analysis, include all 4 LLMs from Figure 6 (not just Vicuna) and include attack models that KDA has not leveraged during training - eg. Direct Request or TAP.
* Extending the "Transfer Attack Via KDA" analysis of Appendix A.1 to a large dataset that was not used in the training of KDA.  - eg. HarmBench (ICLR '24) and on at least one LLM from a different model family to what KDA was fine-tuned on - eg. Mistral.

2. Discuss potential approaches or future work to address KDA's limitation of heavily relying on the presence of successful attack prompts. Eg. Exploring methods to generate novel attack prompts or improve unsuccessful attacks.

3. Please provide a brief comparison of your RedTeam-10K dataset with other existing datasets, including Anthropic's HH-RLHF, highlighting key differences in size, content, and purpose.

**Details Of Ethics Concerns:**

This work has dual use - ie. can be leveraged by attackers for malicious use.

---

### Note · Authors · 2024-11-26

I have read and agree with the venue's withdrawal policy on behalf of myself and my co-authors.